# Protective effects of *Colla Corii Asini* Collagen Peptides on D-galactose injection combined with UVB irradiation-induced aging in mice

Qingdi Luo[1‡], Song Zhang[1,2‡], Zhuo Sun[1], Zhihao Wang[1], Qiulin Yue[1,3], Xin Sun[1,4], Li Tian[1], Baojun Li[3], Kunlun Li[3], Chen Zhao[3], Lin Zhao[1,2,4]*, Le Su[1,2,5]*

1 State Key Laboratory of Biobased Material and Green Papermaking, School of Bioengineering, Shandong Academy of Sciences, Qilu University of Technology, Jinan, China, 2 Shandong Jicui Biotechnology Co., Ltd., Jinan, P.R. China, 3 Jinan Hangchen Biotechnoogy Co., Ltd., Jinan, P.R. China, 4 Shandong Chenzhang Biotechnology Co., Ltd., Jinan, P.R. China, 5 Shengsheng Xiangrong Biotechnology (Shandong) Co., Ltd., Jinan, P.R. China

‡ QL and SZ contributed equally to this work and share first authorship.
* iahb205@163.com (LZ); sule@sdu.edu.cn (LS)

**Data Availability Statement:** All relevant data are within the paper.

**Funding:** This work was supported by Technological Small and Medium sized Enterprise

## Abstract

Skin aging, autonomic mobility, memory function and physical deterioration are important features of aging, and effective anti-aging treatments are important in slowing down these processes. The objective of this research was to evaluate the protective effect of *Colla Corii Asini* (Ejiao) Collagen Peptides (CCACPs) on D-galactose (D-gal) injection combined with UV irradiation-induced senescence in mice. BY-HEALTH collagen oral solution (Bcos) was used as a positive control. Behavioural experiments showed that CCACPs significantly improved voluntary activity, learning memory and exercise endurance in aging mice. Elisa results showed that CCACPs reduced the levels of matrix metalloproteinase-1 (MMP-1) and MMP-3 in the skin, acetylcholinesterase (AChE) in the brain, and alanine aminotransferase (ALT) and azelaic aminotransferase (AST) in the liver of mice, while increasing the levels of collagen I in the skin and SOD in the brain. RT-qPCR revealed that CCACPs reduced the expression of p16, p19 and p21 genes in the liver and hippocampus, as well as the expression of IL-6 in the skin. Histological analysis of brain hippocampus, liver and skin confirmed the protective effects of CCACPs. The findings indicated that CCACPs may potentially slow the aging effects caused by D-galactose and UVB exposure in mice by reducing cellular senescence and oxidative stress levels. The results of this research provide the scientific basis for continuing to advance the extraction of collagen peptides from *Colla Corii Asini* as a potential anti-aging therapy.

## 1. Introduction

Aging is a natural physiological process in which an organism's physiological functions deteriorate over time, leading to increased morbidity and a shorter lifespan [1]. Although the exact mechanisms of aging are unclear [2], studies have shown that aging is a complex process

Innovation Ability Enhancement Project (2022TSGC1076, 2023TSGC0167, 2023TSGC0169), Yantai Development Zone Science and Technology Leading Talents Project (grant number 2020CXRC4, Lin Zhao, and Orlando Borrás - Hidalgo), Key innovation Project of Qilu University of Technology (Shandong Academy of Sciences) (2022JBZ01-06), Spring Industry Leader Talent Support Plan (grant number 2092021033), "Hai You Famous Masters" Industry Leading Talent Support Project (grant number 1362022058).

**Competing interests:** The authors have declared that no competing interests exist.

influenced by multiple factors, including changes in gene expression, external factors, and lifestyle influences [3]. Briefly, problems such as accumulation of DNA damage, oxidative stress, increased inflammation, and disturbed cellular metabolism have all been suggested as important contributors to aging [3]. Delaying aging is critical to improving an individual's fitness and prolonging life [4–6], but cost-effective anti-aging dietary supplements are still limited.

Major risk factors for a range of diseases are strongly linked to aging, including neurodegenerative diseases [7, 8], cardiovascular disease [9, 10], type 2 diabetes [11, 12], and chronic inflammation [13, 14]. These conditions are often driven by mechanisms such as oxidative stress, chronic inflammation, and cellular senescence, all of which are key features of the aging process itself. As such, interventions targeting these underlying mechanisms to slow aging hold great potential for mitigating the onset and progression of aging-related diseases.

In recent years, researchers have been intensively investigating interventions that can slow down aging and reduce the risk of aging-related diseases [15]. Among these interventions, several traditional Chinese medicines have been found to possess significant anti-aging properties, e.g., reducing oxidative stress and delaying cellular aging, and have attracted great interest as potential candidates for new anti-aging strategies [16–20]. *Colla Corii Asini*, a gelatin extracted from donkey skin and concentrated by stewing, is a traditional Chinese medicine known and widely used in traditional Chinese medicine for its blood tonic and anti-fatigue properties [21]. Modern studies have shown that *Colla Corii Asini* is rich in biological activities, such as promoting haemoglobin synthesis and enhancing immune function [21]. *Colla Corii Asini* is rich in biologically active components such as proteins, amino acids, minerals, etc., and its pharmacological effects have been well established [22]. The anti-aging effects of *Colla Corii Asini* were recognised in China a long time ago and are now gradually being scientifically verified. Nevertheless, studies on the anti-aging effects of *Colla Corii Asini* are still limited. Therefore, the aim of this study was to investigate the anti-aging effects of Agaricus aurantium on aging mice.

D-galactose injections combined with ultraviolet B (UVB) radiation are commonly used to establish animal models of senescence [23]. D-galactose accelerates the aging process by inducing oxidative stress and inflammation, whereas UVB radiation mimics the effects of external factors on senescence by directly damaging DNA and exacerbating oxidative stress [24]. This model successfully simulates essential aspects of the aging process in humans and offers a significant resource for researching anti-aging techniques.

While several studies have examined the potential anti-aging benefits of CCACPs on the D-gal model, comprehensive research focusing on an integrated aging model induced by both D-galactose and UVB irradiation remains lacking. Consequently, this study is designed to investigate the protective effects of CCACPs on an integrated aging model in mice, with particular attention to the liver, skin, and hippocampal regions. Key biomarkers in these tissues, including MMP-1, MMP-3, collagen I, AChE, SOD, ALT, and AST, are analyzed. This investigation aims to thoroughly evaluate the anti-aging potential of CCACPs and to provide a scientific foundation for the development of novel anti-aging therapeutic strategies.

## 2. Materials and methods

### 2.1 Materials and chemicals

CCACPs were provided by Jinan Hangchen Biotechnoogy Co., Ltd.. D-galactose was purchased from Shanghai Macklin Biochemical Co., Ltd (Shanghai China). Hydrochloric acid, formic acid and phenol were purchased from Jinan Dingguo Biotechnology Co., Ltd..

## 2.2 Analysis of amino acid composition and molecular weight distribution of CCACPs

For the analysis of the amino acid composition of CCACPs, according to a previous report [25], Amino acid content was determined using a fully automated amino acid analyzer (L-8900 Hitachi Scientific Instruments (Beijing) Ltd.).

The formula is as follows:

$$X = \frac{c \times \frac{1}{50} \times F \times V \times M}{m \times 10^9}$$

X-amino acid content of the sample (g/100g).
c-Amino acid content (nmol/L) in the test solution.
F-sample dilution.
V-Constant volume of the sample after hydrolysis (mL).
M- Molecular weight of amino acid.
m- Mass of sample (g).

To determine the molecular weight distribution of the CCACPs, the samples were dissolved in a 0.1% formic acid solution and centrifuged to obtain a supernatant. Analyses were performed using an Agilent 1260 Infinity multi-detector GPC/SEC system (Agilent 1260 GPC/SEC MDS). The resulting data was then retrieved and analysed for molecular weight distribution using Agilent GPC/SEC software.

## 2.3 Animal testing

All animal experiments were reviewed and approved in accordance with the ARRIVE guidelines for animal research and were conducted in accordance with the UK Animals (Scientific Procedures) Act 1986 and related guidelines, the EU Directive on Animal Experiments 2010/63/EU, the NIH Guidelines for the Care and Use of Laboratory Animals (NIH Publication No. 8023, Revised 1978), and China's Ministry of Health Animal Management Rules (No. 552001) were conducted. The study was approved by the Ethics Committee for Animal Experiments of Qilu University of Technology (Jinan, China) (No. SWS20231104). This study did not require humane endpoints because the animals were humanely euthanized at the end of the experimental period. There were no unexpected deaths, and euthanasia procedures are planned and carried out ethically to ensure humane treatment of all animals.

Twenty-five 7-week-old specific pathogen-free (SPF) male C57BL/6J mice were obtained by Charles River Laboratory Animal Technology (Beijing, China) and housed in a room with a 12-hour light/dark cycle at a temperature of 25 ± 3°C and humidity of 50 ± 5%. After one week of acclimatization feeding, twenty-five mice were randomly divided into five groups (n = 5 for each group): Control group, Model group (D-gal injection and UVB irradiation), positive control group (Bcos from BYHEALTH Co., Ltd), 6 mL / kg), low-dose group of *Colla Corii Asini* Collagen Peptides (L-CCACPs, 67 mg / kg) and high-dose group of *Colla Corii Asini* Collagen Peptides (H-CCACPs, 200 mg / kg). All groups, apart from the Control group, received subcutaneous injections of D-gal (1g/kg) dissolved in 0.9% saline solution into the posterior neck, and were exposed to UVB radiation (50 mJ/cm² for 40 minutes daily) over an 8-week period. Mice belonging to the Bcos, L-CCACPs and H-CCACPs groups were administered treatment via gavage, whereas the Control group mice were given saline through the same method. The body weights of all subjects were consistently monitored on a weekly basis and dosage adjustments were made as necessary. The UVB light was sourced from a UVB-GS-801A lamp (GOLDVISS, China), and radiation intensity was assessed with a HEE47SO24 UV irradiometer (Hangzhou Deep Color Technology Co., Ltd., China).Before UV irradiation, the

dorsal fur of each mouse was shaved with an electric razor, followed by the application of depilatory cream. The distance from the light source to the back of the animals was maintained at 25 cm. During UVB exposure, the mice were housed in groups within the irradiation chamber, allowing free movement [25, 26]. Prior to exposure, the irradiation intensity was calibrated to ensure a consistent dose of 40 mJ/cm$^2$. Throughout the irradiation process, the mice's range of motion was kept within the designated exposure area.

Animal health and behavior were monitored daily by trained staff to ensure well-being. No unexpected deaths occurred during the study. Upon conclusion of the experiment, the mice were humanely euthanized using isoflurane anesthesia and make every effort to minimize the pain, in compliance with institutional and ethical guidelines. Blood was collected and centrifuged at 3000 rpm for 10 minutes to obtain serum. Additionally, the brain, hippocampus, liver, skin, and spleen tissues were harvested from all mice and preserved at -80˚C for subsequent analysis. All animal welfare considerations, including housing with environmental enrichment, were adhered to throughout the experiment. No analgesics were needed during the course of the study, and euthanasia followed institutional guidelines to minimize any potential suffering.

## 2.4 Determination of organ index and body weight

Fresh blood, hippocampus, liver and skin tissues obtained at the end of the experiment were subjected to biochemical assays and histopathological determinations. Liver, spleen and brain were isolated from the body and weighed to calculate the organ index coefficient. The organ index of the animals was calculated as: coefficient (mg/g) = organ weight (mg) / body weight (g).

## 2.5 Behavioral assessments

**2.5.1 Autonomous activity test.** The autonomous activity test was administered prior to all behavioural tests. The device consists of an empty box with a square opening and a capture system for recording trajectories. The mice were placed in the autonomous activity box and the activity of the mice was recorded for a period of 5 minutes.

**2.5.2 Step-down test.** Mice were placed on a jumpable platform, acclimatised for 120 seconds and then given a 0.7mA current stimulus for 180 seconds. When the mice were stimulated with current, they jumped to the safety platform. The time of the first jump was the latency period. If the mice did not jump off the platform, the latency period was calculated to be 3 minutes.

**2.5.3 Swimming experiment.** The mice were placed in the box to swim, and the experiment was stopped when the mice's heads were about to be submerged in water and the swimming time was recorded.

## 2.6 Biochemical analysis

Mice organ fragments (skin, liver, brain, approximately 200 mg each) were homogenized in 2,000 μL of refrigerated saline. Following the homogenization process, the mixture was centrifuged at 4,000 g for 10 minutes at 4˚C, and the resulting supernatant was transferred to separate tubes. ALT and AST levels in liver (E-BC-K236-M, Elabscience Biotechnology Co., Ltd, Wuhan, China), SOD and AChE levels in brain tissue (CSB-E17521m, CUSABIO, Wuhan, China) and collagen I, MMP-1, and MMP-3 levels in skin (CSB-E07925, CSB-E07417m, CSB-E04678m, CUSABIO, Wuhan, China).

## 2.7 Histological studies

Three mice were chosen from each group for H&E staining of liver and hippocampal tissues. The tissues were then removed, fixed, and embedded in 4% paraformaldehyde. The H&E section staining process involved deparaffinizing the sections using an environmentally friendly solution and rehydrating them with different concentrations of ethanol and distilled water. After staining with hematoxylin, the sections were differentiated, washed, and counterstained with eosin. Finally, the sections were dehydrated, stained with eosin, and sealed.

For the Masson staining procedure, the tissue sections were first immersed in Masson A solution overnight. After that, they were rinsed with running water and then placed in a mixture of Masson B and Masson C solution in equal parts for 1 minute. Following another rinse under running water, the slices were stained with Masson D solution for 6 minutes. Next, the sections were washed and subsequently stained with Masson E solution for 1 minute. They were then briefly stained in Masson F solution for approximately 2–30 seconds. To differentiate the slices, a 1% acetic acid rinse was applied. The sections were then dehydrated using anhydrous ethanol and sealed transparently using Masson A-F staining solutions (from Wuhan Servicebio Biotechnology Co.). Pathological changes were examined under a light microscope (Nikon Optical Instruments Co., Ltd., Japan), and the photographs were analyzed using ImageJ software (developed by the US National Institutes of Health, Bethesda).

## 2.8 RNA extraction and Reverse Transcription-quantitative Polymerase Chain Reaction (RT-qPCR)

The extraction of total RNA from the liver and hippocampus of mice was carried out with Trizol reagent (Thermo Fisher Scientific Co., Ltd., USA). Subsequently, the total RNA obtained was transcribed into BcosNA utilizing a reverse transcription kit (ABclone Co., Ltd., Wuhan, China). Real-time fluorescence quantitative PCR analyses were performed using an RT-qPCR kit (ABclone Co. Ltd., Wuhan, China), and all reactions were conducted on a BIO-RAD Real-Time PCR instrument (QIAGEN Hilden, Germany). A three-step PCR program was employed: initial denaturation at 95˚C for 5 minutes; followed by denaturation at 95˚C for 30 seconds, annealing at 58˚C for 30 seconds, and extension at 72˚C for 30 seconds for 40 cycles; and a final extension at 72˚C for 5 minutes. The internal standardization was based on glyceraldehyde-3-phosphate dehydrogenase (GAPDH). The target gene expression levels were determined using the 2-ΔΔCt method. The primer sequences for RT-qPCR are provided in Table 1.

## 2.9 Statistical analysis

All data and results were compared using a one-way ANOVA. Behavioral test data are presented as mean ± SEM, while other data are shown as mean ± SD. Statistical analyses were conducted in GraphPad Prism 9 software, $p < 0.05$ was considered a significant difference.

**Table 1. Primers used in RT-qPCR.**

| Genes | Forward | Antisense |
|---|---|---|
| P16 | 5′-CTCAGCCCGCCTTTTTCTTC-3′ | 5′-CGCCTTCGCTCAGTTTCTCATG-3′ |
| P19 | 5′-GTGGCTCTCGCTACTCTGTTG-3′ | 5′-ATAGTGGATACCGGTGGACCT-3′ |
| P21 | 5′-ACTACCAGCTGTGGGGTGAG-3′ | 5′-TCGGACATCACCAGGATTGG-3′ |
| IL-6 | 5′-CTGCAAGAGACTTCCATCCAG-3′ | 5′-AGTGGTATAGACAGGTCTGTTGG-3′ |
| GADPH | 5′-AACGGATTTGGTCGTATTGGG-3′ | 5′-TCGCTCCTGGAAGATGGTGAT-3′ |

## 3. Results

### 3.1 Amino acid composition of CCACPs

The amino acid composition of the CCACPs is shown in Table 2. The CCACPs contained the highest amount of glycine and glutamic acid, 19.85% and 10.56%, respectively, which is in agreement with previous findings [22] and exhibits the typical amino acid composition characteristic of collagen. This finding suggested that these two amino acids may have important biological activities in combating oxidative stress and inflammatory processes [27, 28]. In addition, CCACPs contains up to 11.77% proline, and relatively high levels of phenylalanine and lysine.

### 3.2 Molecular weight distribution of CCACPs

The distribution of molecular weight indicates the extent of collagen breakdown. Predominantly, the molecular weights of the CCACPs were under 500 Da (Fig 1(A)), comprising approximately 55.76% of total collagen hydrolysates. This implies that nearly all CCACPs consisted of small peptides.

### 3.3 Body weight, organ index and skin moisture

After week 8, a notable reduction in body weight was observed in the Model group compared to the Control group, as depicted in Fig 1(B). The other treated groups exhibited a reversal of the weight loss induced by the Model group. The liver, brain, and spleen indices in the Model group were significantly lower than those in the Control group, as indicated in Table 3. However, an increase in liver, brain, and spleen indices was observed in the Bcos and CCACPs groups compared to the Model-induced aging group. Liver, brain, and spleen indices were initially lower in the Model group but increased after CCACPs treatment.

A 12.68% decrease in skin moisture was noted in the Model group compared to the Control group (Table 3). Following oral administration of CCACPs, the skin moisture content in mice was significantly higher than in the Model group. Differences in skin moisture content between the L-CCACPs and H-CCACPs groups were significant and correlated with the CCACPs dosage. Skin moisture content in the H-CCACPs group exceeded that in the Control group. The study demonstrated a substantial increase in skin water content post-CCACPs ingestion, containing 11.77% proline, potentially stimulating collagen synthesis.

Table 2. The amino acid composition of *Colla Corii Asini* Collagen Peptides.

| Amino Acid Types | Content (%) | Amino Acid Types | Content (%) |
|---|---|---|---|
| Asp | 5.81 | Ile | 1.44 |
| Thr | 2.18 | Leu | 3.21 |
| Ser | 3.94 | Tyr | 1.14 |
| Glu | 10.56 | Phe | 13.26 |
| Gly | 19.85 | Lys | 5.26 |
| Ala | 8.18 | His | 5.42 |
| Cys | 0.00 | Arg | 4.74 |
| Val | 2.68 | Pro | 11.77 |
| Met | 0.56 | - | - |

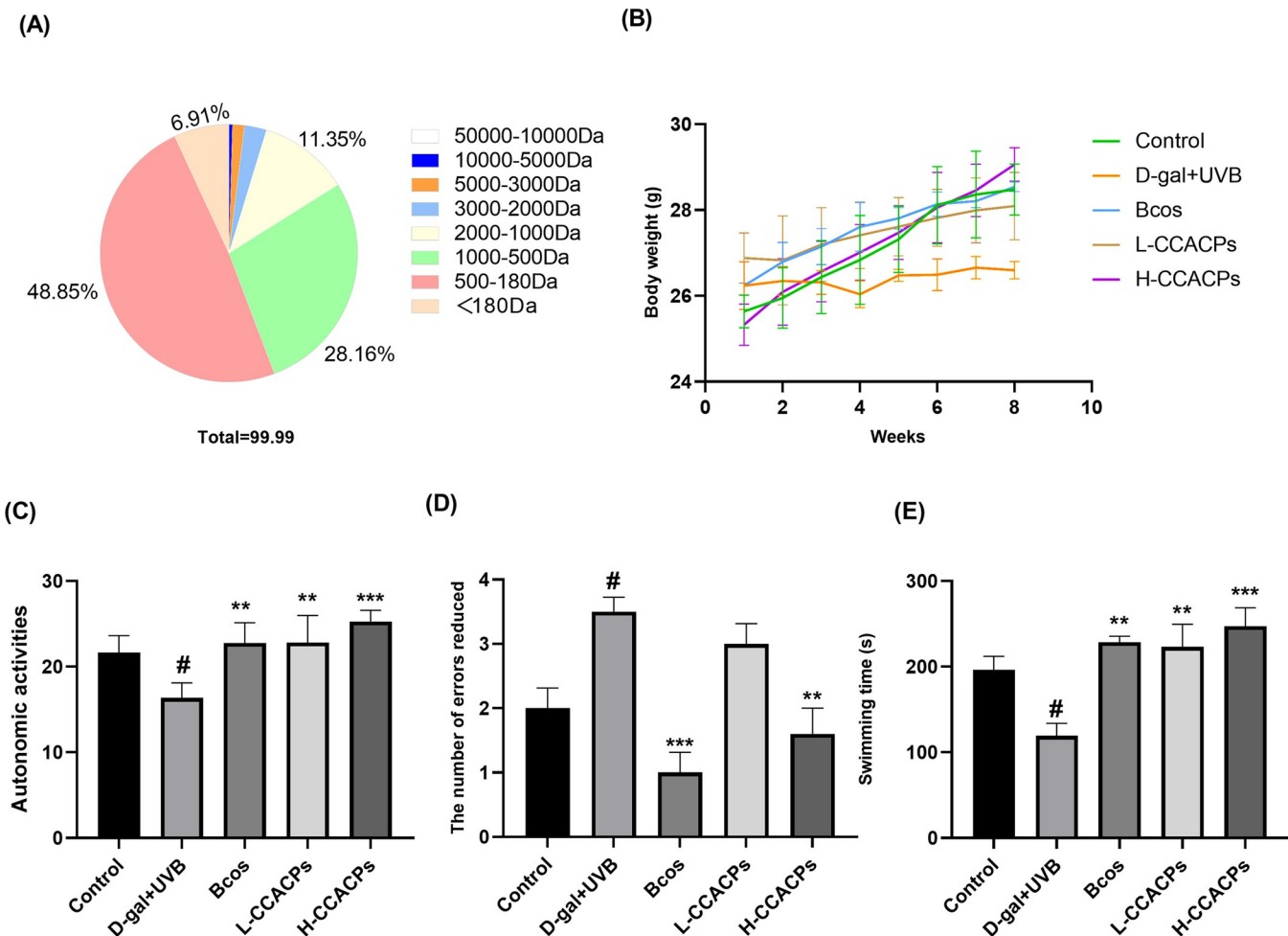

**Fig 1. Distribution of molecular weight of CCACPs and changes in body weight after D-galactose injection combined with ultraviolet irradiation in mice and its improvement of memory cognition and physical performance in mice.** (A) The molecular weight distribution of CCACPs. (B) Body weight of mice. (C) Effects of CCACPs on performance in tests of autonomous activity test in mice. (D) Effect of CCACPs on the performance of mice in the step-down test. (E) Effects of CCACPs on physical performance in mice in the swimming test. (*$p < 0.05$, **$p < 0.01$, ***$p < 0.001$, in relation to D-gal injection combined with UVB irradiation controls; #$p < 0.05$, in relation to control group).

### 3.4 Effects of CCACPs on voluntary activity, memory cognition and physical performance in aging mice induced by D-gal combined with UVB irradiation

In D-gal combined with UVB irradiation-induced aging mice, mice treated with Bcos and CCACPs showed significantly higher activity levels in the autonomous activity test (Fig 1(C)).

The step-down test is a passive avoidance task in which the number of errors is its main index. As shown in Fig 1(D), for the Model group, the number of errors was significantly

**Table 3. Organ indices and skin moisture in mice.**

| Items | Control | Model | Bcos | L-CCACPs | H-CCACPs |
|---|---|---|---|---|---|
| Liver index (mg/g) | 4.31±0.91 | 3.92±1.23 | 4.74±1.42 | 4.61±1.05 | 4.27±1.10 |
| Brain index (mg/g) | 1.51±0.08 | 1.49±0.09 | 1.59±0.08 | 1.51±0.07 | 1.48±0.08 |
| Spleen index (mg/g) | 0.30±0.11 | 0.28±0.13 | 0.31±0.12 | 0.33±0.12 | 0.28±0.11 |
| Skin moisture (%) | 71.10±1.29 | 62.00±1.37 | 76.62±1.54 | 69.02±1.82 | 75.83±1.47 |

higher than the other groups. Compared with the Model group, the Bcos and CCACPs groups could significantly reduce the number of errors in mice.

The role of CCACPs in the physical strength of mice was determined by measuring the swimming duration of mice. Mice in the Model group had reduced swimming duration compared to the Control group (Fig 1(E)). However, treatment with Bcos and CCACPs significantly reversed the Model group. This implies that CCACPs can improve the physical strength of D-gal injection combined with UVB irradiation-induced aging mice.

## 3.5 Measurement of inflammatory factors in serum and skin tissues

IL-6 in the serum of Model group of mice was significantly elevated compared to the Control group (Fig 2(A)). In contrast, IL-6 levels were significantly lower after CCACPs administration, which was significantly different from the Model group, probably because CCACPs have anti-inflammatory effects and can effectively reduce inflammatory responses induced by D-galactose injection and UVB irradiation.

To assess the effect of CCACPs on pro-inflammatory cytokine expression in aging skin, the level of IL-6 was measured. As shown in Fig 2(B), D-gal injection and UVB irradiation resulted

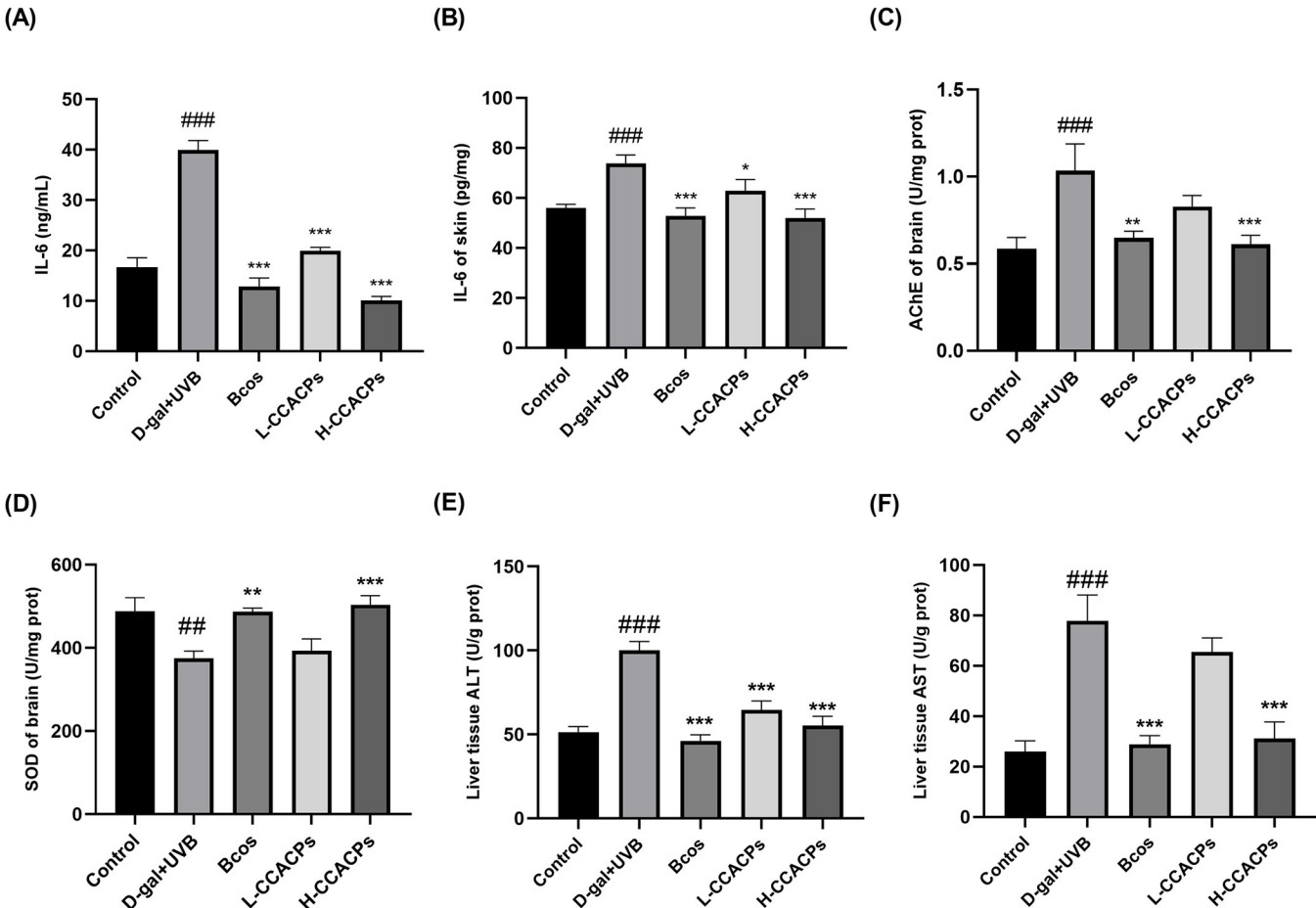

**Fig 2. Effect of CCACPs on inflammatory factors in serum, skin and AChE, SOD in brain and ALT and AST in liver of mice irradiated by D-gal injection combined with UVB.** (A) Serum levels of IL-6 in mice. (B) Levels of IL-6 in mice skin. (C) Levels of AChE in mice brain. (D) Levels of SOD in mice brain. (E) Levels of ALT in mice liver. (F) Levels of AST in mice liver. (*$p < 0.05$, **$p < 0.01$, ***$p < 0.001$, in relation to D-gal injection combined with UVB irradiation controls; ##$p < 0.01$, ###$p < 0.001$, in relation to control group).

in a significant increase in IL-6 levels, whereas CCACPs treatment significantly decreased IL-6 levels in a dose-dependent manner.

### 3.6 Effects of CCACPs on AChE and SOD in the brain

In this study, we determined the effects of CCACPs on AChE and SOD in the brains of mice (Fig 2(C), 2(D)). Long-term injection of D-gal significantly increased AChE levels in the brains of mice (Fig 2C), whereas after a period of treatment, brain AChE activity was significantly reduced and returned to normal levels in the Bcos and H-CCACPs groups. This result suggested that CCACPs may delay brain aging by improving the function of the cholinergic system. The SOD level was significantly reduced in the Model group, however, when the mice were treated with CCACPs, the SOD level in the brains of the mice gradually returned to normal. These results suggested that CCACPs treatment can significantly improve the brain's resistance to oxidative stress.

### 3.7 Effects of CCACPs on ALT and AST in the liver

ALT and AST are reliable indicators for evaluating liver function impairment. As illustrated in (Fig 2(E), 2(F)), levels of AST and ALT were notably increased in the Model group in contrast to the Control group, indicating that D-gal stimulation resulted in liver damage. Conversely, treatment with Bcos and H-CCACPs led to a marked decrease in ALT and AST levels in the aging mice's livers. These findings suggested that H-CCACPs provided a protective role for the liver.

### 3.8 Effect of CCACPs on collagen III, MMP-1, and MMP-3 in skin

As shown in Fig 3, the levels of MMP-1 and MMP-3 in the skin of the Model group were increased by 117.39% and 88.78%, respectively, compared with the Control group. In contrast, Collagen III levels decreased by 28.57%. After oral administration of CCACPs, MMP-1 and MMP-3 decreased by 68% and 53.32%, respectively, in the H-CCACPs treated group compared with the Model group, whereas Collagen III levels increased by 65%. These data suggested that H- CCACPs can mediate protection against aging stress induced by combined UVB and D-galactose treatment by decreasing the expression of MMP-1 and MMP-3 and concomitantly increasing the expression of Collagen III in mice skin.

### 3.9 Protective effects of CCACPs on aging hippocampus, liver and mice skin

In the Model group, as depicted in (Fig 3(D)), hippocampal neurons exhibited disorganization and a noticeable decrease in cell count. Conversely, following CCACPs intervention, there was a marked increase in well-organized neurons and fibers. The morphological composition post-CCACPs treatment closely resembled that of both the blank and positive Control groups, indicating a potential reduction in brain tissue pathology severity.

Histological analysis shown in Fig 3(E) showed that D-galactose injection caused moderate inflammatory cell infiltration and increased cellular space in the liver. In contrast, the administration of CCACPs demonstrated a strong protective effect, effectively mitigating the liver's pathological alterations. The morphological structure of the H-CCACPs group closely resembled that of the Bcos and Control groups.

As shown in Fig 4(A), skin exposure to UVB and injection of D-gal resulted in epidermal thickening, and thus epidermal thickness was significantly increased in the Model group, whereas administration of CCACPs inhibited UVB exposure and increased epidermal

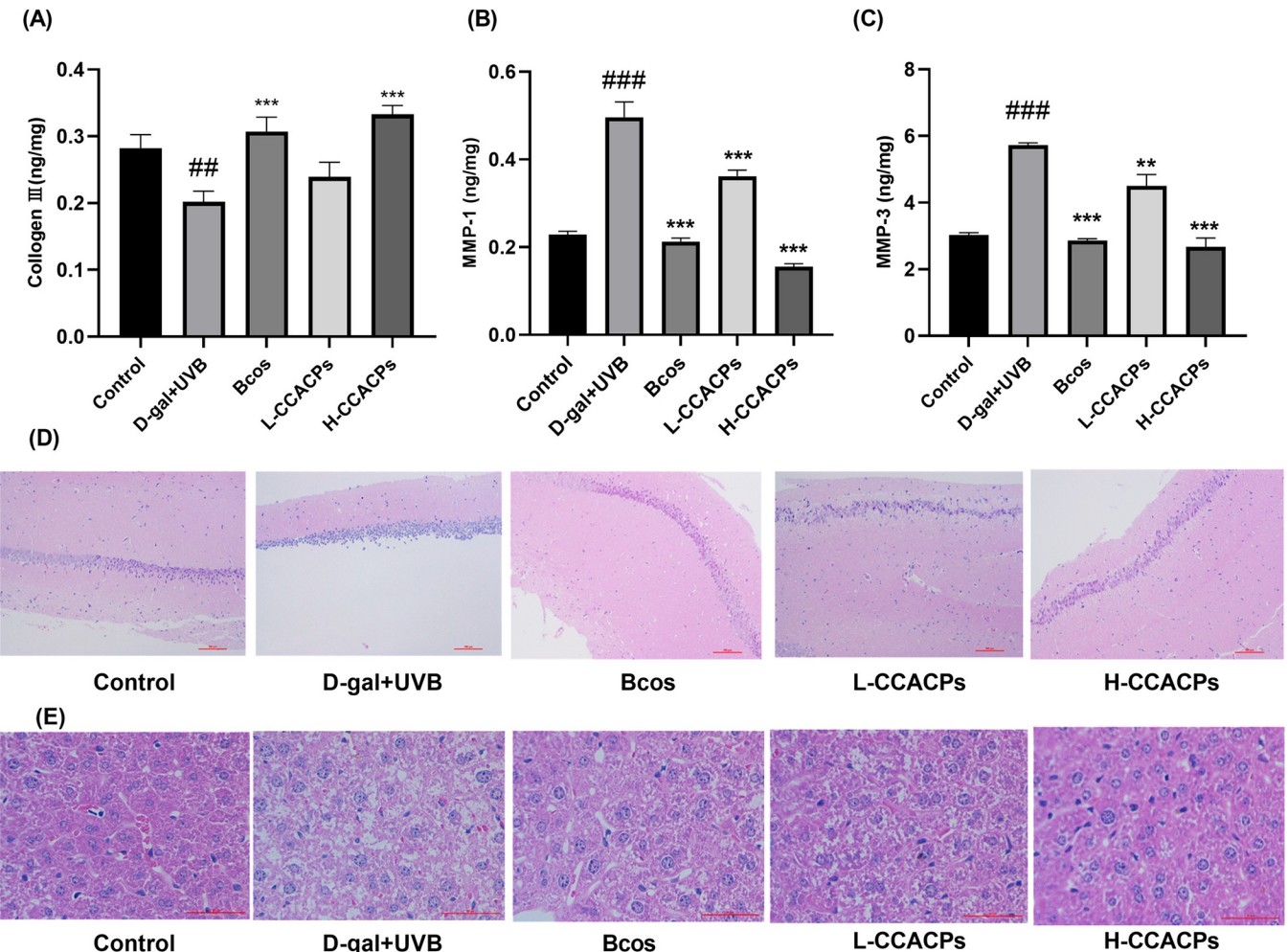

**Fig 3. Effects of CCACPs on type III collagen, MMP-1 and MMP-3 in the skin of D-galactose injections combined with ultraviolet irradiation in mice, and on hippocampal and liver damage.** (A) Levels of Collogen Ⅲ in mice skin. (B) Levels of MMP-1 in mice skin. (C) Levels of MMP-3 in mice skin. (D) Effects of CCACPs on the brain's hippocampus (H&E 10×). (E) Effects of CCACPs on the liver (H&E 40×), (*$p < 0.05$, **$p < 0.01$, ***$p < 0.001$, in relation to D-gal injection combined with UVB irradiation controls; ##$p < 0.01$, ###$p < 0.001$, in relation to control group).

thickness. As shown in Fig 4(B), the area colored blue presented a certain amount of collagen in the dermis of the skin. According to the results of quantification of collagen strength by Image Pro Plus software, UVB irradiation and D-gal injection led to collagen breakdown in the dermis, so the relative collagen strength of the Model group was lower than that of the Control group, but the oral administration of CCACPs prevented the reduction of collagen caused by UVB irradiation and D-gal injection.

### 3.10 Effect of CCACPs on the expression of p16, p19 and p21 genes in the liver and hippocampus and IL-6 gene in the skin of aging mice induced by D-gal injection combined with UVB irradiation

To verify the involvement of CCACPs in averting cellular aging, we evaluated the expression levels of p16, p19, and p21 genes in both the liver and hippocampus. Our findings indicated a significant upregulation of p16, p19, and p21 gene expression in the liver and hippocampus of the Model group when compared to the Control group. In contrast, when compared to the

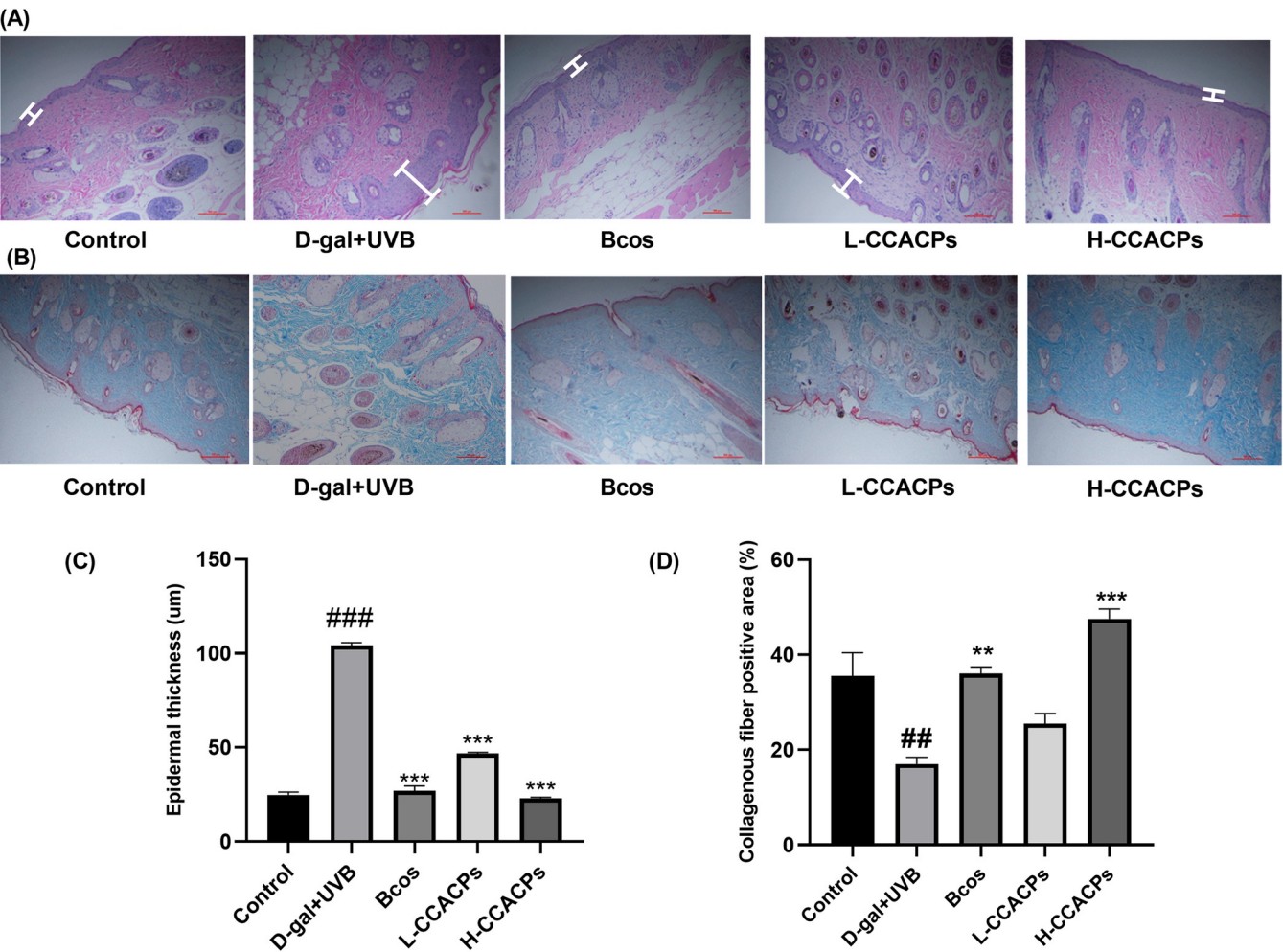

**Fig 4. Effects of CCACPs on skin damage induced by D-gal injection combined with UVB irradiation in aging mice (H&E 10×, Masson 10×).** (A) Sections of paraffin blocks were subjected to H&E staining, and the white line indicates epidermal thickness. Scale bar = 100 μm and dermal thickness was calculated by Image Pro Plus software. (B) Sections of paraffin blocks were stained with Masson trichrome staining to color the collagen in the dermis blue, scale bar = 100 μm, and the relative collagen intensity was calculated by Image Pro Plus software. (C) Measurement and analysis of dermal thickness in each group of mice. (D) Quantification of positive areas of collagen in mice skin. (*$p < 0.05$, **$p < 0.01$, ***$p < 0.001$, in relation to D-gal injection combined with UVB irradiation controls; ##$p < 0.01$, ###$p < 0.001$, in relation to control group).

Model group that was exposed to D-gal injection combined with UVB irradiation, the treatment with Bcos and CCACPs markedly reduced the expression of p16, p19, and p21 genes in the hippocampus and liver (Fig 5(B), 5(C)). Following this, we assessed IL-6 gene expression in the skin. The expression of the IL-6 gene in the skin notably increased as a result of D-gal injection and UVB irradiation in comparison to the Control group, however, the CCACPs treatment group showed a reduction in IL-6 gene expression in the skin (Fig 5(A)).

## 4. Discussion

This research examined the efficacy of CCACPs in aging mice utilizing a model induced by UVB irradiation combined with D-gal injection. The induction of aging by D-gal is a widely accepted model of neurotoxicity stemming from the excessive buildup of ROS and AGEs [23]. Previous research has indicated that oxidative stress can trigger mitochondrial apoptosis induced by ROS, leading to cellular and tissue harm [29]. We used D-gal injection combined

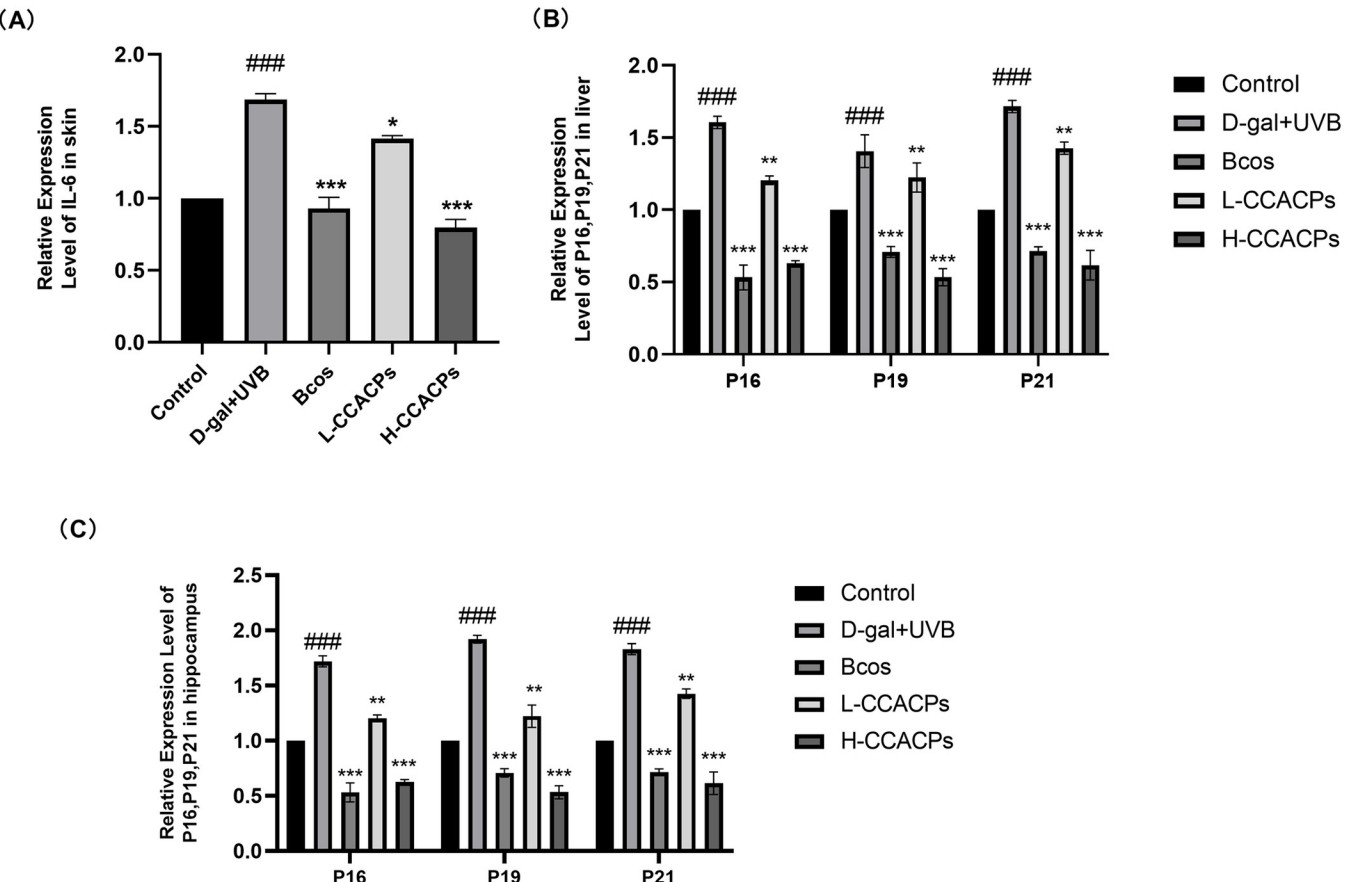

**Fig 5. Effects of CCACPs on IL-6, P16, P19 and P21 gene levels in aging mice induced by D-gal injection and UVB radiation.** (A) Effects of CCACPs on IL-6 gene levels in the skin of aging mice. (B) Effects of CCACPs on P16, P19, P21 gene levels in the liver of aging mice. (C) Effects of CCACPs on P16, P19, P21 gene levels in the hippocampus of aging mice. ($^*p < 0.05$, $^{**}p < 0.01$, $^{***}p < 0.001$, in relation to D-gal injection combined with UVB irradiation controls; $^{###}p < 0.001$, in relation to control group).

with UVB irradiation to establish a mice aging model, which simulates the aging process of mice in the natural state and is closer to the physiological aging of human beings, thus making our results more valuable for practical application. The combined use of D-galactose and UVB irradiation integrates the dual mechanisms of intrinsic and extrinsic aging. D-galactose induces oxidative stress and inflammation to mimic systemic aging, while UVB accelerates skin aging through DNA damage and extracellular matrix degradation [23, 24]. This dual stressor model more closely mirrors the complexity of human aging and provides a clear advantage over studies that use only one of these stressors, such as D-galactose alone [22] or UVB alone [26]. By capturing both systemic and local aging pathways, this approach allows for a more comprehensive assessment of interventions like CCACPs.

As more and more people are focusing on slowing down aging, oral dietary supplement therapeutics are becoming a very attractive strategy for maintaining and ameliorating aging [30]. In order to address this demand, the current study explored the impacts of oral CCACPs on aging. According to previous reports [22], CCACPs exhibit notable antioxidant properties, particularly demonstrating optimal effects in DPPH and ABTS free radical scavenging tests. The antioxidative capabilities aid in the mitigation of oxidative stress and the prevention of free radical-induced harm, ultimately safeguarding cells and tissues against the progression of aging [31]. Glutamic acid plays a crucial role in supporting neurological functions and

maintaining metabolic equilibrium [32]. Glycine and proline are pivotal in the synthesis of human collagen and elastin [33]. Phenylalanine and lysine are essential for facilitating structural protein synthesis and promoting tissue repair, crucial for overall liver function [34–36]. According to our determination of the amino acids of CCACPs and their molecular weights, CCACPs contain large amounts of glycine and glutamic acid, contain high levels of proline, phenylalanine and lysine, and most of the molecular weights of CCACPs are concentrated in the range of 180–500 Da. The amino acid composition of CCACPs was compared with collagen peptides derived from chicken bone and bovine bone in previous studies [37, 38]. CCACPs showed significantly higher levels of phenylalanine (13.26%) and glutamic acid (10.56%) compared to both chicken and bovine collagen peptides, which are known to support tissue repair, structural protein synthesis, and metabolic balance. While the glycine content in CCACPs (19.85%) was slightly lower than that in chicken bone collagen peptides (27.86%), it was comparable to bovine bone collagen peptides (19.84%), ensuring sufficient structural integrity. Proline levels were similar across all collagen sources. These compositional differences underscore the unique amino acid profile of CCACPs, which may contribute to their enhanced bioavailability and systemic anti-aging effects, distinguishing them from other collagen sources that primarily benefit skin health.

According to previous studies, it has been found that collagen peptides demonstrating reduced molecular weights exhibit enhanced biological functions. For instance, in a study by Hongdong [39], it was noted that chub skin collagen peptides with lower molecular weight (200–1000 Da, 65%) displayed superior efficacy in the restoration of UV-induced skin aging in mice when juxtaposed with similar peptides of elevated molecular weight (>1000 Da, 72%). In addition, since CCACPs are mostly composed of small molecules less than 500 Da, this contributes to their better absorption and utilization in vivo. Our study demonstrated that D-gal injections in combination with UVB irradiation resulted in aging-related changes, including slowed body weight gain, a significant decrease in organ indices, and a reduction in skin water content. However, CCACPs were able to partially reverse these adverse effects by promoting weight gain and increasing organ index and skin water content, which may be related to the higher content of glycine and glutamic acid in CCACPs and their smaller molecular weight.

Our research utilized a variety of behavioral experiments to thoroughly evaluate memory, cognitive function, and physical capabilities in elderly mice. The autonomous activity test allowed us to gauge the cognitive and exploratory skills of the mice based on their distance covered and movement patterns in the central area [40]. Normally, animals exhibit proactive exploration behaviors in novel environments, while those with cognitive deficiencies tend to be less active in such settings. Our findings indicate that mice treated with CCACPs displayed heightened independent activities, indicating that CCACPs administration could enhance excitability and adaptation to new environments in aging mice induced by D-gal and UVB irradiation. Aging is a major contributor to cognitive decline characterized by progressive learning and memory deterioration, cognitive dysfunction and conditions like Alzheimer's disease [41]. We utilized step-down test to analyze the impact of CCACPs on memory deficits and cognitive impairment. Our data revealed a significant increase in errors in the mice stepdown test following D-gal injection. However, treatment with CCACPs notably enhanced memory performance, as evidenced by a reduction in errors. Age-related declines in physical performance may lead to increased susceptibility to fatigue [3]. The effect of CCACPs on antifatigue was tested by means of a swimming test, i.e., the effect of CCACPs on the enhancement of physical performance was measured. The findings revealed a significant difference in swimming duration between the Control group and the Model group, indicating that prolonged D-gal injections (8 weeks) resulted in decreased swimming time. In contrast, oral CCACPs administration improved the swimming performance of aging mice induced by D-gal,

demonstrating the potential of CCACPs to enhance physical fitness (Fig 1(E)). The toxicity of the tested substances was assessed using the liver index. The spleen, a key component of both humoral and cellular immunity, is commonly evaluated through the spleen index to gauge immune system functionality. Meanwhile, the brain index is utilized to evaluate neurodegenerative function. Given the link between fatigue and immune system irregularities, the analysis of spleen coefficients showed a reduction in the D-gal-treated group, which was subsequently improved after CCACPs oral administration. This suggests that CCACPs may exert anti-fatigue effects by modulating immune organs, thereby enhancing physical performance.

D-gal injection has been reported to cause liver injury and dysfunction, which increases the levels or activity of certain serum enzymes [22]. The findings of this research illustrate that mice injected with D-galactose exhibited significantly higher levels of ALT and AST in their livers. Additionally, CCACPs notably decreased hepatic levels of ALT and AST. The cholinergic system plays a crucial role in synaptic connectivity and serves as the foundation for transmitting information. Impairment of the cholinergic system is responsible for deficits in memory and cognitive function [41]. The current investigation demonstrated that D-galactose led to an increase in AChE activity in the brain of mice, mirroring cholinergic dysfunction. Conversely, CCACPs hindered AChE activity in the D-galactose-induced mice model, suggesting that CCACPs enhance cognitive abilities by restraining cholinesterase activity. SOD enzymes in the brain are vital for scavenging harmful free radicals and safeguarding neurons against oxidative stress. Insufficient SOD enzyme activity in brain tissue usually leads to increased oxidative damage, which in turn affects neuronal function and activity. This study found that oral administration of CCACPs increased SOD enzyme activity in a D-gal mice model.

Traditional research on *Colla Corii Asini* focuses more on its efficacy and mechanism in slowing down the aging of the organism [22]. In this paper, for the first time, we investigated the effect of CCACPs on the skin anti-aging and organism anti-aging in mice through the D-gal and UVB model, and preliminarily analysed its mode of action. It provides some guidance for revealing the role of *Colla Corii Asini* and its collagen in regulating organism metabolism and organism-skin metabolism correlation research [42]. Collagen III plays an important role in maintaining the elasticity and structure of the skin. Decrease in collagen III in the skin usually leads to decreased skin elasticity and increased aging manifestations [43]. Our study found that treatment with CCACPs increased collagen III levels in aging mice, thereby improving skin elasticity and structure. MMPs in the skin play an important role in collagen degradation. The activity of MMPs in the skin increases after UV irradiation of the skin, and excessive activity of MMPs in the skin usually leads to excessive breakdown of collagen, which in turn causes skin laxity and wrinkle formation [44]. Our study showed that the activities of MMP-1 and MMP-3 in the skin of mice in the Model group were significantly higher than those in the Control group, whereas the treatment with CCACPs reduced the activities of MMPs in the skin of mice, thereby slowing down skin aging.

Histopathologic sections showed that CCACPs protected against senescence in the liver, brain hippocampus, and skin of mice. According to previous reports, aging cells gather in various body tissues as individuals age or encounter diseases [1]. The process of cellular aging is intricately linked to age-related traits, and a decline in aging cells prolongs tissue function and increases lifespan. The accumulation of aging cells improves liver fat buildup and hepatic steatosis, and a treatment regimen combining dasatinib and quercetin decreases aging cells, thereby reducing hepatic steatosis holistically [45]. Exposure to specific environmental contaminants such as paraquat exacerbates the build-up of ageing cells in the brain, which leads to neurodegeneration and reduces protection against paraquat-induced neurological damage [46]. These findings proposed that aging cell management is a growing target for diseases

related to aging, and that hindering aging cells can delay the aging process. Consequently, we observed inflammatory cell infiltration within liver tissue and documented the arrangement of hippocampal neurons. The changes in liver and cerebral hippocampus tissue due to aging were alleviated by the administration of CCACPs, indicating that CCACPs have a safeguarding effect on the liver and brain of mice exposed to D-gal injection along with UVB radiation-induced aging.

p19/p21 and p16 are two major signaling pathways involved in cellular aging. Elevated levels of p21 and p16 expression have been observed in fibroblasts and melanocytes in aging individuals and mice [47]. The accumulation of p16 (increase in p16Ink4a-positive cells in adulthood) has been linked to a reduction in healthy lifespan and an acceleration of age-related changes in various organs. Furthermore, the elimination of p16Ink4a-positive aging cells has been shown to delay the onset of age-related diseases, suggesting that p16 gene expression could potentially serve as a marker for physiological age [48]. Studies have indicated that oxidative stress induced by D-gal leads to DNA damage, telomere shortening, activation of p19/p21, and upregulation of p16 expression through the p38-MAPK pathway [47, 49]. Activation of p16 and p19/p21 contributes to the process of cellular aging. To evaluate the potential of CCACPs in preventing cellular aging, we assessed the expression of p16, p19, and p21 genes in the liver and hippocampus. Our findings revealed that administration of D-gal and UVB radiation resulted in increased expression of p16, p19, and p21 genes in these tissues. However, treatment with CCACPs significantly mitigated these alterations. Additionally, we investigated the expression of the IL-6 gene in the skin. It is well known that D-gal injection and UVB irradiation cause oxidative stress in the skin, which is exacerbated by an excessive increase in inflammatory factors, creating a vicious cycle that leads to further skin damage and aging. As shown in Fig 5(A), compared to the Control group, D-gal injection and UVB radiation caused a marked elevation in IL-6 levels, which was effectively suppressed by CCACPs in the skin. These results suggest that the anti-aging properties of CCACPs may be attributed to their ability to inhibit cellular senescence and oxidative stress.

## 5. Conclusion

In this study, the potential of CCACPs in anti-aging was verified by establishing a mouse aging model with D-galactose combined with UVB irradiation. The results showed that CCACPs not only improved active mobility, memory, physical performance, SOD content in brain tissue and promoted skin collagen synthesis in the mice, but also effectively reduced the levels of matrix metalloproteinase in skin and brain tissue, and AST and ALT in liver tissue. These findings preliminarily validate the application prospect of CCACPs in the field of anti-aging and provide a scientific basis for the future development of CCACPs as an anti-aging therapeutic strategy.

## Supporting information

**S1 Graphical abstract.**
(TIF)

## Author Contributions

**Conceptualization:** Qingdi Luo.

**Data curation:** Qingdi Luo, Song Zhang, Zhuo Sun, Zhihao Wang.

**Formal analysis:** Qingdi Luo, Song Zhang.

**Funding acquisition:** Baojun Li, Kunlun Li, Lin Zhao.

**Investigation:** Qingdi Luo.

**Methodology:** Qingdi Luo.

**Project administration:** Lin Zhao, Le Su.

**Software:** Qingdi Luo, Song Zhang, Zhuo Sun, Zhihao Wang.

**Supervision:** Song Zhang, Qiulin Yue, Xin Sun, Li Tian, Chen Zhao.

**Validation:** Le Su.

**Visualization:** Song Zhang, Le Su.

**Writing – original draft:** Qingdi Luo.

**Writing – review & editing:** Qingdi Luo, Lin Zhao, Le Su.

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
