## [Decision Letter · Decision Letter 0]

9 Dec 2024

PONE-D-24-41650Protective effects of Colla Corii Asini Collagen Peptides on D-galactose injection combined with UVB irradiation-induced aging in micePLOS ONE

Dear Dr. Zhao,

Thank you for submitting your manuscript to PLOS ONE. After careful consideration, we feel that it has merit but does not fully meet PLOS ONE’s publication criteria as it currently stands. Therefore, we invite you to submit a revised version of the manuscript that addresses the points raised during the review process.

We look forward to receiving your revised manuscript.

Kind regards,

Arumugam Muthuvel

Academic Editor

PLOS ONE

Journal Requirements:

Additional Editor Comments:

The authors are asked to revise their manuscript as per the reviewers comment carefully for further process.

Reviewers' comments:

Reviewer's Responses to Questions

**Comments to the Author**

1. Is the manuscript technically sound, and do the data support the conclusions?

Reviewer #1: Yes

Reviewer #2: Partly

2. Has the statistical analysis been performed appropriately and rigorously? 

Reviewer #1: Yes

Reviewer #2: Yes

3. Have the authors made all data underlying the findings in their manuscript fully available?

Reviewer #1: Yes

Reviewer #2: Yes

4. Is the manuscript presented in an intelligible fashion and written in standard English?

Reviewer #1: Yes

Reviewer #2: Yes

5. Review Comments to the Author

Reviewer #1: In this study, the authors have investigated the protective role of Colla Corii Asini collagen peptides on D-Galactose injection combined with UVB irradiation-induced aging in mice. The study is interesting and may appeal to wider audience.

Comments

1. Authors must explain, how they determined low dose(67mg/kg) and high dose (200mg/kg) of CCACPs

2. Figure 1(C), what is the unit of measurement for Autonomic activity? or define the scale

3. In all the bar graphs, authors can move the significant stars from control group to aging model group to better represents the significant changes in model group compared to control group.

4. In 3.10. Indicate the appropriate figure number in the text, instead for just mentioning Figure 5. Mention figure number for Il-6 data

5. Typo

Figure 3, (C) Is it MMP-1 or MMP-3?

Figure 2 (C) axis label (U/mg prot) instead of (U/mgprot)

Materials and methods, 2.3 Animal testing, line 9. Reframe the sentence “There is no expected unexpected

deaths”

Reviewer #2: Explain in detail the uniqueness of employing a combined D-galactose and UVB-induced aging paradigm, setting it apart from other research that used just one of these stressors.

Give further background information on how the results might apply to aging-related diseases in people.

Explain the rationale behind the 8-week exposure period. Would a longer time frame yield more thorough insights?

Give further information about the methods used to regulate the UVB exposure so that each individual experienced the same level of intensity.

To contextualize the peculiarity of CCACP, update Table 2 to include a comparison of its amino acid content with that of other collagen sources.

And additional if some references are added from latest research if available it will be nice

6. PLOS authors have the option to publish the peer review history of their article (what does this mean?). If published, this will include your full peer review and any attached files.

Reviewer #1: No

Reviewer #2: No

---

## [Author Response · Author response to Decision Letter 0]

17 Dec 2024

Dear Editors and Reviewers,

Thank you for your valuable comments and suggestions on our manuscript titled “Protective effects of Colla Corii Asini Collagen Peptides on D-galactose injection combined with UVB irradiation-induced aging in mice” (ID: PONE-D-24-41650). The comments provided have been very valuable for revising and improving our manuscript, and they have been of significant guidance to our research. We have carefully reviewed the comments and made the necessary corrections, hoping for approval. The revised sections are highlighted in the updated manuscript. The main corrections and our responses to the editor and reviewers' comments are as follows:

Reviewer #1: In this study, the authors have investigated the protective role of Colla Corii Asini collagen peptides on D-Galactose injection combined with UVB irradiation-induced aging in mice. The study is interesting and may appeal to wider audience. 1. Authors must explain, how they determined low dose (67mg/kg) and high dose (200mg/kg) of CCACPs.

We greatly appreciated your professional review of our manuscript. The low dose (67 mg/kg) and high dose (200 mg/kg) of CCACPs were determined based on a comprehensive literature review and our preliminary human studies. To begin with, previous studies have reported the use of collagen peptides in animal models at doses ranging from 50 to 800 mg/kg [1–3]. Moreover, we conducted preliminary human studies to identify effective and well-tolerated doses of CCACPs (data not shown). Based on the effective doses observed in these human studies, we applied standard body surface area–based conversion methods to translate the human doses into equivalent animal doses, thereby establishing the appropriate dosing for the mouse experiments. As a result, we selected 67 mg/kg as the low dose and 200 mg/kg as the high dose, ensuring that these doses were both within the range reported in the literature and represented the most effective doses of CCACPs.

2. Figure 1(C), what is the unit of measurement for Autonomic activity? Or define the scale.

The "Autonomic activities" in Figure 1(C) are represented on a normalized scale without explicit units, as defined by the proprietary algorithm of the Animal Behavior Video Analysis System, Ver 20.0703. This value is calculated based on a combination of metrics, including total travel distance (mm), time spent active (s), rest time (s), and average speed (mm/s). The scale reflects the relative activity level of the animal, with higher values indicating more frequent or intense movements. Similar normalization methods have been applied in other studies to combine multiple behavioral metrics to assess animal activity levels [4]. Thus, the values presented in Figure 1(C) are intended to provide a comparative measure of autonomic activity rather than an absolute unit of measurement.

3. In all the bar graphs, authors can move the significant stars from control group to aging model group to better represent the significant changes in the model group compared to control group.

Thanks for your kind suggestion. Following the reviewer’s suggestion, we have revised the bar graphs and their captions accordingly. The significant stars have been moved from the control group to the aging model group in the revised figures to more accurately represent the significant differences in the aging model compared to the control group (Figures 1-5).

4. In 3.10. Indicate the appropriate figure number in the text, instead of just mentioning Figure 5. Mention figure number for IL-6 data.

In response to the reviewer’s suggestion, we have revised the text in Section 3.10 to include specific figure numbers where the relevant data are presented. For the IL-6 expression data, we have explicitly referenced Figure 5(A). Similarly, for the p16, p19, and p21 gene expression data, we have clarified the references to Figure 5(B-C) (Page 22, lines 21-22 and Page 23, lines 3 in the revised manuscript).

5. Typo: Figure 3, (C) Is it MMP-1 or MMP-3? Figure 2 (C) axis label (U/mg prot) instead of (U/mgprot). Materials and methods, 2.3 Animal testing, line 9. Reframe the sentence “There is no expected unexpected deaths”.

a) MMP-1 or MMP-3: We have corrected this typo in Figure 3(C) and confirmed that the correct marker is MMP-3. Figure 3(C) in the manuscript has been revised (Page 20, line 5).

b) Axis label: We have updated the axis label in Figure 2(C) to "U/mg prot" to ensure proper spacing (Page 17, Figure 2(C)).

c) Sentence rephrasing: The sentence in the Materials and Methods section has been rephrased for clarity. The revised sentence is now: "There were no unexpected deaths" (Section 2.3, Page 7, line 6).

Reviewer #2:

1. Explain in detail the uniqueness of employing a combined D-galactose and UVB-induced aging paradigm, setting it apart from other research that used just one of these stressors.

We greatly appreciated your professional and thoughtful review of our manuscript. The combined use of D-galactose and UVB irradiation integrates both intrinsic and extrinsic aging mechanisms. D-galactose induces oxidative stress and inflammation, mimicking systemic aging, while UVB accelerates skin aging through DNA damage and degradation of the extracellular matrix [5,6]. This dual stressor model more accurately reflects the complexity of human aging and provides a more comprehensive approach compared to studies that use only one of these stressors, such as D-galactose [7] or UVB [8]). By integrating both systemic and local aging pathways, this approach provides a more comprehensive assessment of interventions like CCACPs. These points have been added to the Discussion section. (Page 24, lines 9-16). 

2. Give further background information on how the results might apply to aging-related diseases in people.

Major risk factors for a range of diseases are strongly linked to aging, including neurodegenerative diseases [9,10], cardiovascular disease [11,12], type 2 diabetes [13,14], and chronic inflammation [3,15]. These conditions are often driven by mechanisms such as oxidative stress, chronic inflammation, and cellular senescence, all of which are key features of the aging process itself. As such, interventions targeting these underlying mechanisms to slow aging hold great potential for mitigating the onset and progression of aging-related diseases. We have incorporated these points into the Introduction (Page 3, lines 20-22 and Page 4, lines 1-4 ).

3. Explain the rationale behind the 8-week exposure period. Would a longer time frame yield more thorough insights?

Thank you for your comments. The 8-week exposure period is consistent with prior studies in this field, which commonly use a similar duration for comparable experiments [1,8,16,17]. Research has shown that this timeframe strikes a balance between capturing meaningful effects and minimizing potential confounding factors. Therefore, we adopted an 8-week exposure period for the mouse model in this study. In future experiments, we plan to extend the exposure period to further explore the long-term effects.

4. Give further information about the methods used to regulate the UVB exposure so that each individual experienced the same level of intensity.

The UVB irradiation experiment was conducted according to previous reports [8,18]. To ensure consistent UVB exposure across all subjects, we carefully controlled the mice's range of motion. Specifically, the mice were confined to areas smaller than the UVB exposure range, ensuring uniform exposure intensity for all individuals. Additionally, prior to irradiation, we calibrated the intensity using a UVB radiometer to maintain a consistent dose of 40 mJ/cm² throughout the study. The specific modifications in the article are as follows: 

Before UV irradiation, the dorsal fur of each mouse was shaved with an electric razor, followed by the application of depilatory cream. The distance from the light source to the back of the animals was maintained at 25 cm. During UVB exposure, the mice were housed in groups within the irradiation chamber, allowing free movement [8,18]. Prior to exposure, the irradiation intensity was calibrated to ensure a consistent dose of 40 mJ/cm². Throughout the irradiation process, the mice’s range of motion was kept within the designated exposure area (Page 8, lines 3-9).

5. To contextualize the peculiarity of CCACP, update Table 2 to include a comparison of its amino acid content with that of other collagen sources.

Thank you for your comments. Since the amino acid data from other collagen sources are not based on our own results, they have not been included in Table 2. However, we have added a comparison of the amino acid compositions from other collagen sources in the Discussion section. 

The amino acid composition of CCACPs was compared with collagen peptides derived from chicken bone and bovine bone in previous studies [1,19]. CCACPs showed significantly higher levels of phenylalanine (13.26%) and glutamic acid (10.56%) compared to both chicken and bovine collagen peptides, which are known to support tissue repair, structural protein synthesis, and metabolic balance. While the glycine content in CCACPs (19.85%) was slightly lower than that in chicken bone collagen peptides (27.86%), it was comparable to bovine bone collagen peptides (19.84%), ensuring sufficient structural integrity. Proline levels were similar across all collagen sources. These compositional differences underscore the unique amino acid profile of CCACPs, which may contribute to their enhanced bioavailability and systemic anti-aging effects, distinguishing them from other collagen sources that primarily benefit skin health (Page 25, lines 10-21). 

6. Additional references from latest research would be helpful.

Following the reviewer’s suggestion, we have replaced several latest references to support our findings, particularly those related to the role of collagen peptides in aging and their potential therapeutic applications. The references that were changed are 20-27 [20–27].

Journal Requirements:

Thank you for your feedback. I have made the necessary revisions to ensure that the manuscript complies with PLOS ONE's style requirements, including the correct file naming conventions. Please feel free to let me know if any further adjustments are needed.

2. Please ensure that you provide the correct grant numbers for the awards you received for your study in the ‘Funding Information’ section.

Thank you for pointing out the inconsistency in the funding information. To address the inconsistency noted in the funding information, upon resubmission, we have provided the correct funding numbers for the awards supporting our research in the 'Funding Information' section. The manuscript has been appropriately revised to ensure consistency with the information provided in the submission system. (Page 31, Line 18-22 and Page 32, Line 1 in the revised manuscript).

We have made all necessary revisions to the manuscript based on your constructive feedback. We believe these changes have significantly improved the manuscript and we hope that the revised version meets the expectations of the reviewers and editors.

Thank you again for your time and helpful suggestions.

Sincerely,

Lin Zhao, Ph.D.

School of bioengineering

Qilu University of Technology 

Jinan, 250353, China

Fax: + 86 531 89631776

Tel: + 86 531 86529099

E-mail: iahb205@163.com

Reference 

1. Song H, Zhang S, Zhang L, Li B. Effect of Orally Administered Collagen Peptides from Bovine Bone on Skin Aging in Chronologically Aged Mice. Nutrients. 2017;9: 1209. doi:10.3390/nu9111209

2. Wang J, Wu Y, Chen Z, Chen Y, Lin Q, Liang Y. Exogenous Bioactive Peptides Have a Potential Therapeutic Role in Delaying Aging in Rodent Models. International Journal of Molecular Sciences. 2022;23: 1421. doi:10.3390/ijms23031421

3. Singh A, Schurman SH, Bektas A, Kaileh M, Roy R, Wilson DM, et al. Aging and Inflammation. Cold Spring Harb Perspect Med. 2024;14: a041197. doi:10.1101/cshperspect.a041197

4. Liu J, Chen D, Wang Z, Chen C, Ning D, Zhao S. Protective effect of walnut on d‐galactose‐induced aging mouse model. Food Sci Nutr. 2019;7: 969–976. doi:10.1002/fsn3.907

5. Azman KF, Zakaria R. d-Galactose-induced accelerated aging model: an overview. Biogerontology. 2019;20: 763–782. doi:10.1007/s10522-019-09837-y

6. Choi S-I, Han H-S, Kim J-M, Park G, Jang Y-P, Shin Y-K, et al. Eisenia bicyclis Extract Repairs UVB-Induced Skin Photoaging In Vitro and In Vivo: Photoprotective Effects. Mar Drugs. 2021;19: 693. doi:10.3390/md19120693

7. Wang D, Liu M, Cao J, Cheng Y, Zhuo C, Xu H, et al. Effect of Colla corii asini (E’jiao) on D-galactose induced aging mice. Biol Pharm Bull. 2012;35: 2128–2132. doi:10.1248/bpb.b12-00238

8. Cui B, Wang Y, Jin J, Yang Z, Guo R, Li X, et al. Resveratrol Treats UVB-Induced Photoaging by Anti-MMP Expression, through Anti-Inflammatory, Antioxidant, and Antiapoptotic Properties, and Treats Photoaging by Upregulating VEGF-B Expression. Oxid Med Cell Longev. 2022;2022: 6037303. doi:10.1155/2022/6037303

9. Guo L, Li X, Gould T, Wang Z-Y, Cao W. T cell aging and Alzheimer’s disease. Front Immunol. 2023;14: 1154699. doi:10.3389/fimmu.2023.1154699

10. Liu R-M. Aging, Cellular Senescence, and Alzheimer’s Disease. Int J Mol Sci. 2022;23: 1989. doi:10.3390/ijms23041989

11. Evans MA, Sano S, Walsh K. Cardiovascular Disease, Aging, and Clonal Hematopoiesis. Annu Rev Pathol. 2020;15: 419–438. doi:10.1146/annurev-pathmechdis-012419-032544

12. Liberale L, Badimon L, Montecucco F, Lüscher TF, Libby P, Camici GG. Inflammation, Aging and Cardiovascular Disease: JACC Review Topic of the Week. J Am Coll Cardiol. 2022;79: 837–847. doi:10.1016/j.jacc.2021.12.017

13. Halim M, Halim A. The effects of inflammation, aging and oxidative stress on the pathogenesis of diabetes mellitus (type 2 diabetes). Diabetes & Metabolic Syndrome: Clinical Research & Reviews. 2019;13: 1165–1172. doi:10.1016/j.dsx.2019.01.040

14. Bellary S, Kyrou I, Brown JE, Bailey CJ. Type 2 diabetes mellitus in older adults: clinical considerations and management. Nat Rev Endocrinol. 2021;17: 534–548. doi:10.1038/s41574-021-00512-2

15. Gulen MF, Samson N, Keller A, Schwabenland M, Liu C, Glück S, et al. cGAS–STING drives ageing-related inflammation and neurodegeneration. Nature. 2023;620: 374–380. doi:10.1038/s41586-023-06373-1

16. Li H, Zhai B, Sun J, Fan Y, Zou J, Cheng J, et al. Antioxidant, Anti-Aging and Organ Protective Effects of Total Saponins from Aralia taibaiensis. Drug Des Devel Ther. 2021;15: 4025–4042. doi:10.2147/DDDT.S330222

17. Huang L, You L, Aziz N, Yu SH, Lee JS, Choung ES, et al. Antiphotoaging and Skin-Protective Activities of Ardisia silvestris Ethanol Extract in Human Keratinocytes. Plants (Basel). 2023;12: 1167. doi:10.3390/plants12051167

18. Zhang Z, Zhu H, Zheng Y, Zhang L, Wang X, Luo Z, et al. The effects and mechanism of collagen peptide and elastin peptide on skin aging induced by D-galactose combined with ultraviolet radiation. Journal of Photochemistry and Photobiology B: Biology. 2020;210: 111964. doi:10.1016/j.jphotobiol.2020.111964

19. Cao C, Xiao Z, Tong H, Liu Y, Wu Y, Ge C. Oral Intake of Chicken Bone Collagen Peptides Anti-Skin Aging in Mice by Regulating Collagen Degradation and Synthesis, Inhibiting Inflammation and Activating Lysosomes. Nutrients. 2022;14: 1622. doi:10.3390/nu14081622

20. Yang F, Yang Y, Xiao D, Kim P, Lee J, Jeon Y-J, et al. Anti-Photoaging Effects of Antioxidant Peptide from Seahorse (Hippocampus abdominalis) in In Vivo and In Vitro Models. Mar Drugs. 2024;22: 471. doi:10.3390/md22100471

21. Kim J, Kim H, Seo W-Y, Lee E, Cho D. Collagen Type VII (COL7A1) as a Longevity Mediator in Caenorhabditis elegans: Anti-Aging Effects on Healthspan Extension and Skin Collagen Synthesis. Biomol Ther (Seoul). 2024;32: 801–811. doi:10.4062/biomolther.2024.127

22. Yang D, Liu Q, Xu Q, Zheng L, Zhang S, Lu S, et al. Effects of collagen hydrolysates on UV-induced photoaging mice: Gly-Pro-Hyp as a potent anti-photoaging peptide. Food Funct. 2024;15: 3008–3022. doi:10.1039/d3fo04949c

23. Lee SG, Ham S, Lee J, Jang Y, Suk J, Lee YI, et al. Evaluation of the anti‐aging e

---

## [Editor Report · Decision Letter 1]

26 Dec 2024

Protective effects of Colla Corii Asini Collagen Peptides on D-galactose injection combined with UVB irradiation-induced aging in mice

PONE-D-24-41650R1

Dear Dr. Zhao,

We’re pleased to inform you that your manuscript has been judged scientifically suitable for publication and will be formally accepted for publication once it meets all outstanding technical requirements.

Kind regards,

Arumugam Muthuvel

Academic Editor

PLOS ONE
---

## [Editor Report · Acceptance letter]

14 Jan 2025

PONE-D-24-41650R1 

PLOS ONE

Dear Dr. Zhao, 

I'm pleased to inform you that your manuscript has been deemed suitable for publication in PLOS ONE. Congratulations! Your manuscript is now being handed over to our production team.

Kind regards, 

on behalf of

Dr. Arumugam Muthuvel 

Academic Editor

PLOS ONE